# Stable Low-Grade Degenerative Spondylolisthesis Does Not Compromise Clinical Outcome of Minimally Invasive Tubular Decompression in Patients with Spinal Stenosis

**DOI:** 10.3390/medicina57111270

**Published:** 2021-11-19

**Authors:** Morsi Khashan, Khalil Salame, Dror Ofir, Zvi Lidar, Gilad J. Regev

**Affiliations:** 1The Department of Neurosurgery, Tel-Aviv Sourasky Medical Center, Tel-Aviv 6423906, Israel; morsi.kh@gmail.com (M.K.); khalils@tlvmc.gov.il (K.S.); drorofir.2y@gmail.com (D.O.); lidar.zvi@gmail.com (Z.L.); 2Sackler Faculty of Medicine, Tel Aviv University, Tel-Aviv 6423906, Israel

**Keywords:** decompression, minimally invasive, spinal stenosis, spondylolisthesis

## Abstract

*Background and Objectives:* In recent literature, the routine addition of arthrodesis to decompression for lumbar spinal stenosis (LSS) with concomitant stable low-grade degenerative spondylolisthesis remains controversial. The purpose of this study is to compare the clinical outcome, complication and re-operation rates following minimally invasive (MIS) tubular decompression without arthrodesis in patients suffering from LSS with or without concomitant stable low-grade degenerative spondylolisthesis. *Materials and Methods:* This study is a retrospective review of prospectively collected data. Ninety-six consecutive patients who underwent elective MIS lumbar decompression with a mean follow-up of 27.5 months were included in the study. The spondylolisthesis (S) group comprised 53 patients who suffered from LSS with stable degenerative spondylolisthesis, and the control (N) group included 43 patients suffering from LSS without spondylolisthesis. Outcome measures included complications and revision surgery rates. Pre- and post-operative visual analog scale (VAS) for both back and leg pain was analyzed, and the Oswestry Disability Index (ODI) was used to evaluate functional outcome. *Results:* The two groups were comparable in most demographic and preoperative variables. VAS for back and leg pain improved significantly following surgery in both groups. Both groups showed significant improvement in their ODI scores, at one and two years postoperatively. The average length of hospital stay was significantly higher in patients with spondylolisthesis (*p*-value< 0.01). There was no significant difference between the groups in terms of post-operative complications rates or re-operation rates. *Conclusions:* Our results indicate that MIS tubular decompression may be an effective and safe procedure for patients suffering from LSS, with or without degenerative stable spondylolisthesis.

## 1. Introduction

The initial treatment of lumbar spinal stenosis (LSS) is usually conservative [1], and surgery is indicated when conservative management fails. Currently, LSS is the most common indication for lumbar spinal surgery [2], and data shows that surgery results in superior outcomes compared to conservative management [3,4,5].

In patients with LSS, decompression procedures can increase instability as result of bony and soft tissue damage. This risk is believed to be higher in patients with preexisting spondylolisthesis [6]; thus, the addition of arthrodesis to decompression was recommended for some of these patients mainly when instability is evident. However, in recent literature, the routine addition of arthrodesis to decompression for LSS with concomitant stable low-grade degenerative spondylolisthesis remains a matter of debate [7,8,9,10,11,12,13].

Minimally invasive (MIS) tubular decompression using a unilateral approach for bilateral decompression is a less invasive approach that has been shown to preserve the midline structures, the contralateral facets, and much of the contralateral lamina, as well as to reduce bony and soft tissue injury [14,15,16]. Yet, MIS decompression was shown to result in comparable long-term outcomes to traditional open spinal decompression procedure [17,18]. Hence, the choice of MIS decompression may be a particularly appealing option for patients with lumbar stenosis and concomitant stable low-grade degenerative spondylolisthesis.

The aim of this study is to compare clinical outcomes and complications of MIS tubular decompression surgery in patients suffering from LSS, with or without concomitant stable low-grade degenerative spondylolisthesis. We hypothesized that the clinical outcomes of patients with LSS and preexisting stable degenerative spondylolisthesis are not inferior to the those of patients suffering from LSS without spondylolisthesis. 

## 2. Materials and Methods

After approval from our local institutional review board in 2013, we established a broad database prospectively collecting multiple medical parameters of all patients undergoing elective spinal procedures. Patients’ consent was not required for this study, due to its retrospective nature. This study is a retrospective reviewed our prospectively collected medical data of patients who underwent one or two levels of lumbar decompression surgery between November 2013 and May 2017. We included patients above the age of 18 years who underwent MIS tubular laminoforaminotomy for LSS with or without stable low-grade degenerative spondylolisthesis. 

The diagnosis of spondylolisthesis and the measurement of the spondylolisthesis slip were based on standing lateral neutral radiographs. The severity of spondylolisthesis was graded according to the Meyerding classification system, based on the ratio of vertebral body slippage relative to the caudal vertebral body [19]. Meyerding grades I and II were considered low-grade, and grades III and IV were considered high-grade. Instability of the operated levels was defined as a ≥3 mm difference between the measured translation in flexion and the extension standing radiographs [20]. 

We included patients suffering from radiculopathy or spinal claudication, after failure of conservative management. All patients that were operated in our spine surgery unit, at this time period, due to spinal stenosis with or without spondylolisthesis were treated by the same MIS technique described. Exclusion criteria included patients with isthmic spondylolisthesis, patients with radiographic instability at the operated level, patients with back pain that is worse than leg pain, or patients who suffer primarily from mechanical back pain suggesting instability. Mechanical back pain was defined as back pain that is worsened with motion without leg pain, suggesting radiculopathy or neurogenic claudication. We also excluded patients who underwent surgeries at more than two spinal levels and patients treated for any other non-degenerative indication. Patients with stable low-grade degenerative spondylolisthesis were assigned to the S group, and patients without spondylolisthesis were assigned to the N group. 

Demographic data and preoperative data, including medical history, medical comorbidities, and American society of anthologists (ASA) scores, were recorded. To simplify the presentation of the medical history data, we classified patients’ comorbidities into eight categories: cardiovascular, hypertension, diabetes, endocrine, neoplastic, neurological, pulmonary, and renal comorbidity. Operative data included the operated spinal level, and incidence of intraoperative complications. 

All procedures were performed in a single medical center by four senior spinal surgeons who are well experienced in MIS surgeries. MIS decompression procedures were carried out routinely under general anesthesia using a 20-mm tubular retractor system (METRx [Medtronic Sofamor Danek, Memphis, TN, USA]) and a surgical microscope. Surgery was performed using a unilateral approach for bilateral decompression, as described previously [21,22].

Outcome measures included the length of hospital stay (LOS), post-surgical complications rates, re-operation rates, and patient-reported outcomes that were collected using a questionnaire in routine outpatient visits. Patient-reported outcomes included the Oswenstry disability index (ODI) and the visual analog scale (VAS) for both back and leg pain, before surgery, and one and two years following the surgery. Clinically significant improvements in disability and pain were defined as any improvement that exceeds the minimum clinically important difference (MCID) for ODI and VAS. We previously used published MCID values as follows: 1.16 for NRS back pain, 1.36 for NRS leg pain, and 12.40 for ODI [23].

## 3. Statistical Analysis

The statistical analysis was performed using SPSS version 19 (IBM Corp., Armonk, NY, USA). Significant differences between the groups were determined using independent samples *t*-test, the X2 test, and the Fisher exact test, to evaluate categorical variables’ independence. Preoperative to postoperative ODI and VAS changes within each group were analyzed with paired *t*-tests. A *p*-value of <0.05 was considered statistically significant. 

## 4. Results

### 4.1. Participants

We used our database to identify all patients who underwent MIS tubular lumbar decompressive procedures between November 2013 and May 2017 (n = 274). After exclusion of MIS discectomy procedures, 128 patients remained. We then excluded patients with less than 12 months follow-up leaving, 96 patients available meeting the inclusion criteria

Ninety-six patients met the inclusion criteria and were assigned to two groups. The study S group included 53 patients, and the control N group included 43 patients. The average follow-up was 27.5 ± 8 months. All patients had follow-up of one year or longer. Seventy three patients (76%) had follow-up of two years or longer.

### 4.2. Outcome Data and Main Results

The average age was 68.9 ± 11.2 years and it was higher in the S group (71.5 ± 8.7 vs. 65.8 ± 13.2, *p*-value = 0.02) (Table 1). The distribution of ASA scores was similar between the two groups. There was no significant difference between the groups in terms of comorbidities (Table 1). 

Length of hospital stay: Forty-two patients (44%) were discharged within 24 h and 67 patients (79%) within 48 h following surgery. The average length of hospital stay was 2.97± 4.6 days and was significantly shorter in the N group (Table 2), Appendix A
Table A1. 

Complications: The overall complication rate was 14.6% and was comparable between the groups (*p*-value = 0.1). Incidental durotomies were recorded in seven patients (7.3%), four patients (9%), and three patients (6%) in the S group and the N group, respectively (*p*-value = 0.7). Neurological complications included motor deficit in one patient (1%) in the N group. This patient had a new postoperative weakness of the tibialis anterior muscle which recovered completely during within the first post-operative week. Wound infections were diagnosed in two patients (2%); one of these patients underwent wound revision surgery, and the other was treated conservatively. One patient (1%) had postoperative pneumonia (N group). Residual stenosis at the index level was found in three patients (3.1%), two of whom underwent revision surgery (one in each group).

Re-operation: Six patients (6.2%) underwent re-operation (three patients (7%) from the N group and three patients (6%) from the S group (*p*-value = 0.56)). In the N group, one patient underwent revision decompression without fusion, 31 months following index surgery due to residual stenosis at the index level. A second patient underwent decompression for to adjacent level stenosis due to disc herniation four months following index surgery. The third patient underwent transforaminal lumbar interbody fusion (TLIF) at the index level, eight months following the surgery due to a lack of improvement in his back and leg pain. In the S group, one patient underwent wound revision due to surgical site infection. A second patient underwent revision decompression at the index level ten months following surgery due to inadequate decompression and residual stenosis. The third patient underwent TLIF at the index level at 18 months postoperatively due to foraminal disc herniation at the operated level. 

Patient-reported outcome: The average baseline ODI scores and VAS for back and leg pain were similar between the groups (Figure 1, Figure 2 and Figure 3, Table 3). At one and two years following the surgery, there was no significant difference between the groups in terms of ODI scores or VAS for back or leg pain. A statistically and significant improvement was found in ODI and VAS for back and leg pain within the whole cohort and within each group after one (*p*-value of <0.01) and two years (*p*-value of <0.01) of follow-up compared to the baseline. This improvement was also clinically significant as the improvement in the ODI and VAS values exceeded the MCID values in the entire cohort and within each group after one and two years of follow-up compared to baseline. The proportion of patients from each group who had improvement exceeding the MCID is demonstrated in Table 4. 

## 5. Discussion

Our results show that MIS decompression may be an effective procedure for LSS. Despite a higher average age in the S group, the two groups were comparable in complications and revision rates as well as in patient-reported outcomes.

Length of hospital stay:

The mean LOS was significantly shorter in the N group (2 ± 1.6 days) compared to the S group (3.1 ± 4.2 days) (*p*-value = 0.01). It should be noted that this difference may be the fact that the average age was higher in the S group. A previous study of our group the compared the outcomes of MIS decompression between older and younger patients, founded that LOS was significantly higher in the older group [24]. However, in the current study, no statistically significant correlation was found between age and the length of stay (*p*-value = 0.18). After controlling for age There was still statistically significant differences between the S and the N groups (*p*-value = 0.05).

Complication rates:

The rate of incidental durotomy in our cohort was 7.3%, and did not differ significantly between the groups. This rate is consistent with the reported rate for open lumbar decompression [25,26] and for minimally invasive decompression procedures of the lumbar spine [27,28]. Surgical site infection was diagnosed in two patients (in the S group). The difference in infection rates between the groups was not statistically significant. The infection rate (2%) of the entire group is consistent with the results of multiple studies that reported a lower risk of infection after minimally invasive procedures as compared to open procedures [29,30]. Moreover, previous reports demonstrated similarly low surgical site infection rates in MIS lumbar decompressive procedures [31,32].

Patient-reported outcome:

As for patient-reported outcomes, the ODI and VAS for back and leg pain improved significantly within the entire cohort, and within each group, after one (*p*-value< 0.01) and two years (*p*-value = <0.01). These results suggest that MIS decompression results in reduced pain and improved function over two years. 

Re-operation rate:

Overall, six patients (6%) underwent revision surgery, of which two patients (2%), one in each group, needed arthrodesis. Three patients were re-operated for residual stenosis (two in the N group and one in the S group), and one patient in the S group underwent wound revision. Revision rates were similar between the groups, and the presence of stable low-grade degenerative spondylolisthesis did not impact the reoperation rates, nor the necessity for further fusion.

The reported rates of reoperation after decompression alone for lumbar spinal stenosis vary widely in the literature with some advantage for MIS decompression over open decompression. Blumenthal et al. reported a reoperation rate of 37.5% following open decompression alone for LSS and grade I degenerative spondylolisthesis in 3.6 years follow-up [33]. In all of their patients, arthrodesis was performed for pain due to instability at the index level. Ahmad et al. reported 17% reoperation rate following open decompression via spinous process osteotomy for LSS and preexisted degrative spondylolisthesis. Of the 17%, 6% required arthrodesis [34]. In a recent meta-analysis investigating patients with LSS and grade I degenerative spondylolisthesis, the rate of reoperation was found to be 8.5% following decompression alone [35]. Using the same surgical technique as in our study, Alimi et al. reported a reoperation rate of 12.9%, of which 3.5% required arthrodesis [28]. They found a revision rate of 16.3% after open decompression and 5.8% after MIS decompression. The secondary fusion rate was 2.8% in open decompression and 3.3% in MIS decompression. The reoperation rate in our study is in accordance with the literature of MIS decompression for LSS and LSS with preexisting low-grade spondylolisthesis [13,17,18,28]. 

The lower rate of reoperation in MIS decompression compared to open decompression may be explained by the difference in the invasiveness of these procedures. Open decompression results in more extensive soft-tissue injury and higher infection rates [29,30], which may influence the revision rate. In open decompression, the longer incision and the more extensive muscle dissection may lead to posterior muscle dysfunction as a result of edema [14], denervation [36], or the reduction in the paravertebral muscle cross-sectional area [37]. Moreover, resection of the spinous process and disruption of the midline structures compromise the posterior tension band effect [38]. Biomechanical studies have found open bilateral laminotomy and MIS laminotomy to be superior to traditional open laminectomy [38,39,40]. A systematic review by Guha et al. [6] concludes that iatrogenic instability following lumbar decompression occurs less commonly with MIS procedures. However, as post-operative flexion/extension radiographs were not available for analysis for this study, we are unable to draw any conclusion with regards to mechanical instability based on our results. The results and conclusion of our studies are merely clinical.

Minimally invasive decompression of LSS:

The literature clearly shows that surgical treatment results in superior outcomes when compared to conservative management [3,4,5]. Yet, the recommended type of surgical intervention is not well-established. Open decompression is an effective and widely accepted treatment for LSS. However, it has some remarkable disadvantages when compared to MIS procedures. In this study, we showed that the clinical outcome of MIS decompression for LSS is comparable to the previously published outcome for open decompression. Our results are also consistent with similar studies that analyzed clinical outcome of MIS decompression [13,24,28,29,31,32]. 

LSS and concomitant degenerative spondylolisthesis:

Whether arthrodesis should be routinely added to decompression for LSS with preexisting spondylolisthesis is still a matter of debate. Some authors advocated adding arthrodesis to prevent post-operative instability [8,9,10]. Other authors showed no benefit in adding arthrodesis to decompression in cases of low-grade spondylolisthesis [11,12,13]. Although decompression without fusion may lead to some slip progression, this does not necessarily affect the clinical outcomes, or increase functional disability, as revealed in a report by Ravinsky et al. on patients undergoing MIS decompression for LSS with spondylolisthesis [41]. It should be also noted that the addition of arthrodesis to decompression may prolong recovery, increase complications, and increase re-operation rates [2,42,43].

We emphasized that our study did not directly address the necessity of arthrodesis in patients with LSS and degenerative spondylolisthesis. However, our results showed that the presence of low-grade spondylolisthesis by itself did not affect the outcome or complications in patients undergoing MIS decompression for LSS. This study indicates that, in the specific subset of patients with LSS and preexisting stable low-grade spondylolisthesis, the outcome of MIS tubular decompression is comparable to that of patients with LSS without spondylolisthesis. These results are in agreement with those of recent studies [11,12,13,35].

There are a few limitations to our study. Patients outcome data did not include routine radiological assessment of the patients. As such, post-operative changes in the degree of spondylolisthesis were not included in our analysis, and no conclusion regarding post-operative alterations in spinal stability could be reached. However, we can conclude that, even if post-operative radiographs showed worsening of the spondylolisthesis in some patients, this has not manifested in their clinical and functional outcomes. Lastly, post-operative follow-up of our patients was limited to 27.5 ± 8 months. It is possible that this fact can explain our low re-operation rate compared to previous studies that were based on longer post-operative follow-up periods [33]. 

## 6. Conclusions

These results indicate that minimally invasive decompression may be an effective and safe procedure for patients with lumbar spinal stenosis. MIS decompression may also be considered in patients with concomitant stable low-grade spondylolisthesis. In these patients, MIS decompression may be considered an alternative to open decompression with or without fusion. 

## Figures and Tables

**Figure 1 medicina-57-01270-f001:**
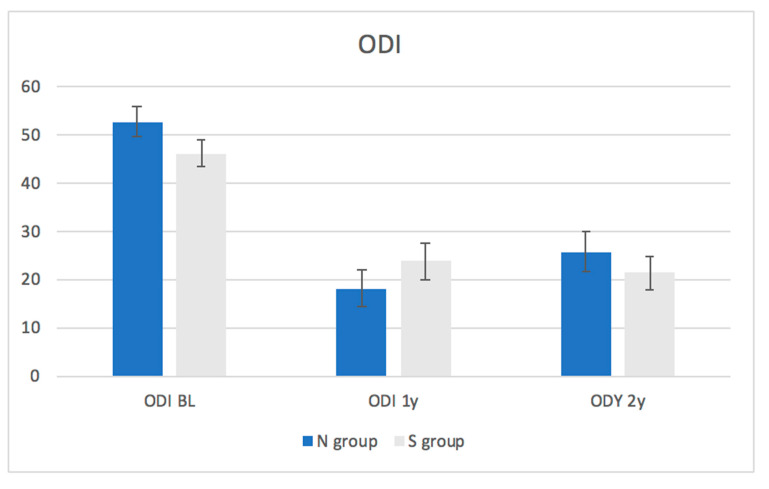
Changes in the Oswestry Disability Index (ODI) within in each group throughout the follow-up period.

**Figure 2 medicina-57-01270-f002:**
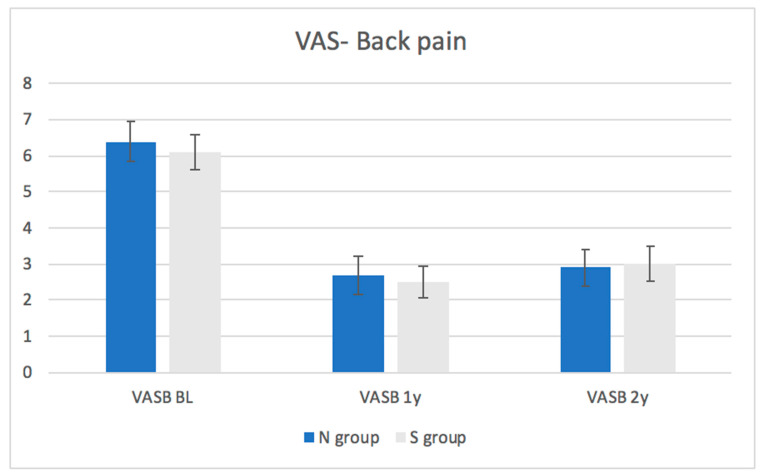
Changes in the visual analogue scale (VAS) for back pain in each group throughout the follow-up period.

**Figure 3 medicina-57-01270-f003:**
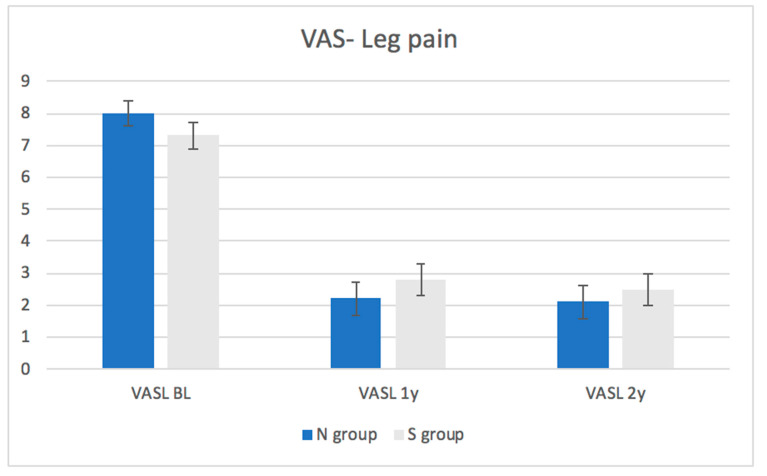
Changes in visual analogue scale for leg pain in each group throughout the follow-up period.

**Table 1 medicina-57-01270-t001:** Demographic variables, basic health status, and comorbidities in both groups.

	N	S	*p*-Value
Total	43	53	
Male	27 (63%)	26 (49%)	0.22
Age	65.8 ± 13.2	71.5 ± 8.7	0.02
BMI	27.4 ± 6.3	28.6 ± 4.5	0.47
BMI > 30	10 (23%)	18 (34%)	0.27
Smoking	11 (26%)	12 (23%)	0.46
ASA score			
I	5 (12%)	3 (6%)	0.49
II	23 (53%)	32 (60%)	
III	13 (30%)	15 (28%)	
IV	2 (5%)	4 (8%)	
Comorbidities			
Cerebrovascular	4 (9%)	4 (8%)	0.52
Renal	4 (9%)	1 (2%)	0.17
Oncological	3 (7%)	3 (6%)	0.56
HTN	23 (53%)	37 (70%)	0.08
DM	13 (30%)	17 (32%)	0.51
Cardiovascular	27 (63%)	31 (58%)	0.68
Other endocrine	9 (21%)	12 (23%)	0.52

Values are mean ± SD, number (%), or as otherwise indicated. ASA—American Society of Anesthesiologists.

**Table 2 medicina-57-01270-t002:** Length of hospital stay, postoperative complications, and revisions in both groups.

	N	S	*p*-Value
Total	43	53	
LOS	2.0 ± 1.6	3.1 ± 4.2	0.01
Complications	9 (21%)	5 (9%)	0.10
Immediate complication	5 (12%)	3 (6%)	0.46
After discharge	3 (7%)	3 (6%)	0.56
Durotomy	4 (9%)	3 (6%)	0.70
Neurological	1 (42%)	0 (0%)	0.45
SSI	0 (0%)	2 (4%)	0.50
Pneumonia	1 (2%)	0 (0%)	0.45
UTI	0 (0%)	0 (0%)	-
PE/DVT	0 (0%)	0 (0%)	-
Residual stenosis	2 (5%)	1 (2%)	0.58
Other complications	0 (0%)	1 (2%)	0.55
Revision	3 (7%)	3 (6%)	0.56

Values are mean ± SD, number (%), or as otherwise indicated. PE/DVT= pulmonary embolism/Deep vein thrombosis.

**Table 3 medicina-57-01270-t003:** Changes in the Oswenstry Disability Index (ODI) on the visual analogue scale (VAS) for back pain within each group throughout the follow-up period.

		Baseline	1 Year	*p*-Value 1y-BL	1 Year	*p*-Value 1y-BL
ODI	All patients	49.1 ± 21.2	21.4 ± 26.1	<0.01	23.4 ± 25.8	<0.01
	N group	52.8 ± 20.2	18.2 ± 24.7	<0.01	25.8 ± 26.9	<0.01
	S group	46.2 ± 21.6	23.9 ± 27.2	<0.01	21.5 ± 24.9	<0.01
VAS- Back pain	All patients	6.2 ± 3.5	2.6 ± 3.2	<0.01	2.9 ± 3.5	<0.01
	N group	6.4 ± 6.4	2.7 ± 2.7	<0.01	2.9 ± 2.9	<0.01
	S group	6.1 ± 3.4	2.6 ± 3.1	<0.01	3.0 ± 3.5	<0.01
VAS- Leg pain	All patients	7.6 ± 2.9	2.5 ± 3.5	<0.01	2.3 ± 3.5	<0.01
	N group	8.0 ± 8.0	2.2 ± 2.2	<0.01	2.1 ± 2.1	<0.01
	S group	7.3 ± 7.3	2.8 ± 2.8	<0.01	2.4 ± 2.4	<0.01

Values are mean ± SD, number.

**Table 4 medicina-57-01270-t004:** Proportions of patients from each group with improvement exceeding the MCID in each of the patient reported outcome measure.

		Total	N	S	*p*-Value
ODI 1 year	Patients with available scores	82	35	47	
	Improvement above the MCID	53 (65%)	27 (77%)	27 (57%)	0.16
ODI 2 year	Patients with available scores	68	30	38	
	Improvement above the MCID	45 (66%)	23 (77%)	22 (58%)	0.12
VASB 1 year	Patients with available scores	80	33	47	
	Improvement above the MCID	44 (55%)	16 (48%)	28 (60%)	0.16
VASB 2 year	Patients with available scores	67	28	39	
	Improvement above the MCID	39 (58%)	14 (50%)	25 (64%)	0.31
VASL 1 year	Patients with available scores	79	33	46	
	Improvement above the MCID	55 (70%)	24 (73%)	31 (67%)	0.8
VASL 2 year	Patients with available scores	64	28	36	
	Improvement above the MCID	41 (64%)	17 (61%)	24 (67%)	0.79

## Data Availability

The data presented in this study are available on request from the corresponding author. The data are not publicly available due to privacy and ethical restrictions.

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
