# Peer review of "Stable Low-Grade Degenerative Spondylolisthesis Does Not Compromise Clinical Outcome of Minimally Invasive Tubular Decompression in Patients with Spinal Stenosis"

_medicina, 2021, doi:10.3390/medicina57111270_

Round 1

Reviewer 1 Report

Stable Low-Grade Degenerative Spondylolisthesis Does Not 2

Compromise Clinical Outcome of Minimally Invasive Tubular 3 Decompression in Patients with Spinal Stenosis.

This is a very well written paper addressing a very common problem. The question of the value of added fusion to decompression has been debated extensively. Still, despite multiple attempts to address that, no consensus has been reached.

The authors conducted a retrospective review of a prospectively collected data. They compared patients suffering from lumbar spinal stenosis with and without low grade spondylolisthesis. Both groups were treated with minimal invasive decompression without fusion. Their long-term follow-up shows comparable outcomes regardless of the existence of the low grade spondylolisthesis.

Overall, I think that this paper is well written and will have an important added value to the common debate of the need to fuse a degenerative lumbar spine.

Author Response

Thank you very much for your positive review.

No additional changes or revisions were proposed by the reviewer.

Reviewer 2 Report

This is a retrospective cohort study that seeks to compare outcomes between of MIS tubular decompression without arthrodesis in patient with lumbar stenosis with spondylolisthesis (S) vs without stable spondylolisthesis (N). 96 patients were included in the study with a mean follow up of 27.5 months. The S group had 53 patients while the (N) group had 43. There was a significant difference in age between the two populations but otherwise no baseline differences. The authors found significant improvement in both groups in VAS and ODI which was maintained at 1 and 2 years. The average LOS in the S group was longer. Otherwise there were no significant differences between the groups in post operative complications and revision rates. Overall I think this is a good study. However I have the following comments

1) It appears that patient follow up is approximately 50% which introduces the potential for bias. Please discuss this in the limitations

2) In your discussion you discuss Blumenthal's paper who reported 37.5% reoperation rate following open decompression at an average of 3.6 years follow up. However, in your study the follow up was approximately 2.3 years, this raises the question of whether with more time the reoperation rate would rise. I would recommend adding some sources that have similar length of follow up to yours if available. Otherwise I would bring it up in the limitations sections

3) In the papers where you discuss open treatment, are these full laminectomies or open midline sparing laminectomies. How does the open midline sparing laminectomy compare to tubular MIS decompression?

Overall very good study with interesting results

Author Response

Thank you for your thorough review and valuable comments. 

In reply to your comments: 

1) It appears that patient follow-up is approximately 50% which introduces the potential for bias. Please discuss this in the limitations

We agree, we added this information to the discussion regarding this study's limitations.

2) In your discussion you discuss Blumenthal's paper who reported 37.5% reoperation rate following open decompression at an average of 3.6 years follow up. However, in your study the follow up was approximately 2.3 years, this raises the question of whether with more time the reoperation rate would rise. I would recommend adding some sources that have similar length of follow up to yours if available. Otherwise I would bring it up in the limitations sections.

We agree, we added this information to the discussion regarding this study's limitations.

3) In the papers where you discuss open treatment, are these full laminectomies or open midline sparing laminectomies. How does the open midline sparing laminectomy compare to tubular MIS decompression?

For the papers discussed here, total laminectomies were performed. As for open cases of decompression without fusion, we are definitely in favor of performing them using a midline structure sparing technique. To the best of our knowledge, there are no studies in the literature that compare midline sparing laminectomy to tubular decompression.  We believe that using this technique provides many of the benefits of MIS decompression regarding the preservation of spinal stability.